# Concurrent validity of novel smartphone-based apps monitoring barbell velocity in powerlifting exercises

Alexander Renner[1]*, Benedikt Mitter[2], Arnold Baca[1]

1 Department of Biomechanics, Kinesiology and Computer Science in Sport, Centre for Sport Science and University Sports, University of Vienna, Vienna, Austria, 2 Research Group for Industrial Software (INSO), Vienna University of Technology, Vienna, Austria

⊕ These authors contributed equally to this work.
* alexander.renner@univie.ac.at

**Data Availability Statement:** All relevant data are available at https://doi.org/10.5281/zenodo.10864547.

**Funding:** The author(s) received no specific funding for this work.

## Abstract

The aim of this study was to determine the validity of three smartphone applications measuring barbell movement velocity in resistance training and comparing them to a commercially available linear transducer. Twenty competitive powerlifters (14 male and 6 female) completed a progressive loading protocol in the squat, bench press and deadlift (sumo or conventional) until reaching 90% of the highest load they had achieved in a recent competition. Mean velocity was concurrently recorded with three smartphone applications: Qwik VBT (QW), Metric VBT (MT), MyLift (ML), and one linear transducer: RepOne (RO). 3D motion capturing (Vicon) was used to calculate specific gold standard trajectory references for the different systems. A total of 589 repetitions were recorded with a mean velocity of (mean ± standard deviation [min-max]) 0.44 ± 0.17 [0.11–1.04] m·s$^{-1}$, of which MT and ML failed to identify 52 and 175 repetitions, respectively. When compared to Vicon, RO and QW consistently delivered valid measurements (standardized mean bias [SMB] = 0 to 0.21, root mean squared error [RMSE] = 0.01 to 0.04m·s$^{-1}$). MT and ML failed to deliver a level of validity comparable to RO (SMB = -0.28 to 0.14, RMSE = 0.04–0.14m·s$^{-1}$), except for MT in the bench press (SMB = 0.07, RMSE = 0.04m·s$^{-1}$). In conclusion, smartphone applications can be as valid as a linear transducer when assessing mean concentric barbell velocity. Out of the smartphone applications included in this investigation, QW delivered the best results.

## Introduction

The term velocity-based training (VBT) in the context of resistance training is used to describe training in which the movement velocity of strength exercises is integrated as a variable into the prescription, monitoring and/or evaluation of the training program [1].

While recent research supports a more in-depth analysis of velocity curves and the use of variables such as the time of the concentric movement phase, the two most commonly addressed VBT variables are mean velocity and peak velocity [1, 2]. Mean velocity refers to the average concentric movement velocity, and peak velocity refers to the maximum

**Competing interests:** The authors have declared that no competing interests exist.

instantaneous velocity reached during the concentric phase of the movement [3, 4]. In the context of powerlifting exercises, mean velocity is usually the variable of choice [5]. This is due to mean velocity being more reliable compared to peak velocity [6–8].

In the past, the popularity of integrating velocity as a training parameter has increased due to better availability of linear transducers (LTs) [9]. This trend seems to be continuing due to more and different measurement systems entering the market. As of the writing of this paper, four different technologies can be distinguished for measuring barbell velocity: accelerometers, LTs, smartphone applications (SAs), and optic devices. While highly valid and reliable measurement systems for measuring barbell velocity are available on the market, the main obstacle for including barbell velocity as a variable for monitoring and prescribing training seems to be the high cost associated with such measurement systems [10]. Therefore, SAs could provide a low-cost alternative to more established technologies, and they may significantly enhance accessibility for practitioners to incorporate mean velocity as a training parameter. Additionally, smartwatch applications have emerged as similar tools that also show promising results in this domain [11].

SAs designed for the assessment of movement velocity typically function according to the same principle of measuring and tracking a reference point of known size in a video and basing all kinematic calculations on this reference. In barbell exercises, for instance, SAs may use weight-plates with standardized diameters as the reference object [12]. The theoretical limitations of such an approach are resolution and frames per second of the recorded video. In theory, this basic principle is straightforward and accurate. However, in practical applications, several factors can influence its accuracy. The most significant of these include the tilting of the barbell, camera distortion, and movement in the coronal plane. Thus, the question arises as to the extent to which SAs achieve valid measurement results in real-world settings. Previous validation studies of SAs that measure barbell velocity have mainly concentrated on the app MyLift, formerly known as PowerLift. These studies have yielded inconclusive results, with some supporting its accuracy and reliability [3, 13–15] while others concluded that it is highly prone to error [16–18]. Another problem which seems to be more specific to SAs are missed repetitions and ghost repetitions. Missed repetitions refer to repetitions which were performed and recorded but not identified as a repetition by the measurement system due to error. Ghost repetitions refer to extra repetitions that were identified by the system, but not performed by the lifter. Ghost repetitions usually occur when the unracking of the barbell is counted as a repetition or when a single repetition is evaluated as multiple repetitions.

Previous work in this field has demonstrated that the validity of different technologies and even of different devices applying the same technology can vary based on which type of exercise is used [10, 19]. However, research has yet to investigate the concurrent accuracy and precision of multiple different SAs developed to assess movement velocity in resistance training. Therefore, the present study was designed to evaluate the validity and missed repetition rate of the smartphone apps Qwik VBT, Metric VBT and MyLift, and compare them to a gold standard 3D Motion Capture System (Vicon). Moreover, a LT (RepOne) was applied as a practical reference, to give practitioners further insight into whether SAs could replace more established and expensive VBT hardware.

## Methods

### Experimental approach to the problem

Following the recommendations of Weakly et al. [10] this study utilized gold standard criterion measures across a range of relative loads and exercises. Therefore, Vicon Nexus 3D motion capture technology was utilized to establish precise trajectory references for the

outputs of each system. The reliability of this technology in measuring 3D kinematics has been previously confirmed elsewhere [20, 21].

To further enable comparison with established field-based technologies, a LT (RepOne Tether, RepOne Strength, New York, United States) was included alongside the 3D motion capture gold standard. Although the current LT model from RepOne (RO) has not been validated yet, the earlier model, known as the "Open Barbell System," was validated against a gold standard 3D high-speed motion capture system. In this study, the Open Barbell System was declared a valid measurement method [22].

Because the average concentric velocity decreases gradually with higher percentages of the one repetition maximum (1RM) [5], a gradual loading protocol was selected to encompass a broader spectrum of velocities. A further benefit of a gradual loading protocol is that sticking points and other changes to barbell kinematics might only become apparent closer to muscular failure [23, 24]. The decrease and subsequent increase of concentric movement velocity throughout the sticking point region imposes a further challenge on the different systems.

As suggested by Weakly et al. [10], the different systems were evaluated across a variety of exercises with distinct kinematic features. In this study, powerlifting exercises were selected, specifically the squat, bench press, and deadlift, each possessing distinct characteristics. These free-motion barbell exercises introduce additional challenges compared to Smith-machine exercises, including horizontal movements that technologies might encounter in real-world situations [15]. The squat is characterized by a long range of motion, a relatively straight bar path due to the fact that the center of mass must be maintained over the base of support and an eccentric/concentric movement pattern [25]. When performed by experienced powerlifting athletes and to the technical standards of the International Powerlifting Federation (IPF) the bench press is characterized by a short range of motion, a J-curve shaped concentric bar path [26], an eccentric/isometric/concentric movement pattern and also different velocity profiles depending on the range of motion [25, 27]. The deadlift is characterized by a range of motion that typically falls between the squat and bench press, and a relatively straight bar path. In contrast to the other investigated exercises, it is initiated by a concentric movement phase [25]. In competitive powerlifting, two major deadlift types can be distinguished: the conventional deadlift, where the hands are placed laterally to the legs, and the sumo deadlift, where the hands are placed between the legs. Due to the wider stance, the sumo deadlift is typically associated with a shorter range of motion compared to the conventional deadlift [28]. It can be assumed that lifters, due to their training history and individual anthropometric constitution, are primarily proficient in one type of deadlift. Since the investigated systems apply the same processing algorithm for the determination of barbell kinematics in the two deadlift types, the participants were instructed to perform the deadlift using the same type as in their most recent competition.

## Subjects

Twenty competitive powerlifters (14 male and 6 female) volunteered to participate in this study. The participants were recruited from the 25th of May 2023 until the 28th of June 2023. Sample characteristics are summarized in Table 1. An extensive questionnaire was used to ensure that all participants met the inclusion criteria. The inclusion criteria were being free of illness and injury and having a valid sports pass of their national powerlifting federation. Furthermore, all participants were required to exhibit participation at a powerlifting competition within the last 6 months prior to the test, for which they achieved a WILKS coefficient of at least 300. The WILKS coefficient pertains to the Powerlifting total, which is the sum of the heaviest successful attempts in the squat, bench press and deadlift. It has been reported to be a

**Table 1. Subject characteristics.**

| Variable | Mean ± SD (min-max) |
|---|---|
| Age (y) | 26±4.9 (18–38) |
| Body mass (kg) | 83.4±19.8 (55.8–147.7) |
| Height (cm) | 173.2±9.6 (160–196) |
| Absolute squat 1RM (kg) | 193.5±54.5 (120–310) |
| Absolute bench press 1RM (kg) | 123.5±44.6 (55–230) |
| Absolute deadlift 1RM (kg) | 224.9±59.4 (142.5–360) |
| Powerlifting total (kg) | 541.9±155.9 (332.5–900) |
| Relative squat 1RM (kg·kg$_{BM}^{-1}$) | 2.32±0.43 (1.68–3.46) |
| Relative bench press 1RM (kg·kg$_{BM}^{-1}$) | 1.46±0.36 (0.88–2.28) |
| Relative deadlift 1RM (kg·kg$_{BM}^{-1}$) | 2.71±0.48 (2.07–3.88) |
| Wilks coefficient | 394.88±58.99 (326.34–499.05) |
| IPF GL coefficient | 80.89±11.61 (66.38–102.86) |

1RM, One-repetition maximum; IPF GL, International Powerlifting Federation Good Lift; kg$_{BM}$, kg body mass; SD, standard deviation.

valid method for adjusting powerlifting performance to body mass and sex [29]. Subjects were briefed on the procedure and purpose of the trial before signing a written informed consent form. The methodological protocol was conducted following the Declaration of Helsinki and approved by a local ethics committee (reference number 00976).

## Procedures

All subjects visited the laboratory on a single occasion. Upon arrival, personal data of the subject was recorded. This included sex, age, self-reported body height, body mass at recent competition and strength performance at recent competition. Based on the results of the recent competition, the loads for the gradual loading protocol were calculated. After a guided warm-up by a certified Level II Coach of the IPF, subjects performed ascending loading protocols for squat, bench press, and deadlift in the specified order. For each exercise, participants completed a single repetition with 45%, 50%, 55%, 60%, 65%, 70%, 75%, 80%, 85% and 90% of their recent competition best, with barbell loads being rounded to the nearest multiple of 2.5kg. To keep the execution of the powerlifting movement standardized, all repetitions were performed according to the technical rules of the IPF [25]. A licensed powerlifting referee was present at every session and enforced the technical rules. For all recorded repetitions, general recommendations for the assessment of force-velocity and load-velocity data were followed as closely as possible [30]. If a lift was unsuccessful or deemed to be too heavy for successful completion by either the subject or the present coach the protocol was stopped prematurely. All exercises were performed using a 20-kg powerlifting barbell, calibrated weight plates, competition collar and an IPF competition combo rack (Eleiko, Halmstad, Sweden). All three SAs request the diameter of the loaded weight plate. To enable the use of 450mm weight plates in all load conditions, 2.5kg technique plates were applied for loads below 65kg.

For each SA, the latest version was installed on one of three identical iPhones SE 2022 (Apple Inc., Cupertino, CA). For Metric VBT (MT), the version 2.3.1 was used. For My Lift (ML), the smartphone application My Jump Lab version 3.2.9 was used. For Qwik (QW), the version 0.94 was used. All three iPhones were placed in a 1m high custom-built smartphone stand with all three iPhones in direct contact with each other. This stand was placed 3m laterally from the lateral end of the barbell. The iPhones were positioned so that the center of the

barbell was in the middle of the screen. To nullify any potential bias related to the specific position (left, middle, or right) of the iPhone, the iPhones were rotated after every subject.

All SAs request the user to film the exercise from the side and adjust the plate diameter if a non-standard sized weight plate (450mm diameter) is used. For MT and ML the videos were recorded with the SAs themself. For QW, the videos were recorded with the native camera application (1920x1080, 60FPS) of the iPhone, and imported into the SA afterwards.

The tether of the RO was placed on the opposing barbell sleeve. The floor unit was aligned for each exercise to minimize tether angle offset from perfect vertical orientation. The RO was connected to an iPhone X (Apple Inc., Cupertino, CA) through Bluetooth running the RepOne Personal app version 1.1.5.

Neither QW, ML, nor RO have the option to select a specific exercise before recording the repetitions. For MT, the correct exercise was chosen before starting the recording. The mean velocity measurements of the RO unit and the SAs were documented during the testing sessions. Wherever possible, the full range of capabilities offered by the individual SAs was used to ensure the highest quality measurements. Specifically, for ML, it was ensured that the "Plate detected" status was displayed before every repetition. For MT, the video function was used to distinguish between ghost repetitions and actual repetitions whenever feasible. The actual repetitions were included in the statistical analysis, while also documenting the ghost repetitions separately. Importantly, these ghost repetitions were not counted as missed repetitions as long as a clear distinction between the ghost repetition and the actual repetition was possible. For QW, the plates were tagged as accurately as possible. Additionally, the native video footage was trimmed to start approximately 5 seconds before the start of each repetition and conclude roughly one second after the bar was returned to the competition combination or placed on the ground.

Reflective pearl markers with a diameter of 26mm (B&L Engineering, Santa Ana, CA) were placed on each lateral end of the barbell. The trajectories of the markers were recorded using a 3D motion tracking system composed of twelve 8.1 megapixel infrared cameras (Vicon V8; Vicon Motion Systems Limited, Oxfordshire, England), sampling at a rate of 200 Hz. The raw trajectory data were then processed in 3D reconstruction software (Vicon Nexus 2.14; Vicon Motion Systems Limited, Oxfordshire, England) by applying a fourth-order Butterworth filter with phase- shift correction and a 10-Hz cutoff frequency to the position. Mean velocity was determined as the first derivative of distance with respect to time during the concentric phase of each repetition. This involved analyzing the velocity data and identifying segments where the velocity surpassed predefined thresholds. The velocity data was processed using a Python script for this purpose. An instantaneous velocity threshold of 0.02m/s was established for rep detection, while a 10cm threshold was set to filter out smaller repetitions resulting from barbell unracking, reracking and slight movements during this process. When both of these thresholds were exceeded, a rep was detected for this specific segment and in these identified segments, mean velocity was calculated by dividing distance by time.

In cases where multiple repetitions were detected within a single performed repetition due to unracking or reracking motions, the repetitions were cross-checked against the 3D reconstructed barbell movement to ensure the correct repetition was selected.

The applied Python script was uploaded to a public repository (https://doi.org/10.5281/zenodo.10864547). The experimental setup is displayed in Fig 1.

## Statistical analysis

Analysis was completed separately for each practical system (RO, QW, MT, ML) and for each investigated exercise (squat, bench press, deadlift) using two different linear mixed effects

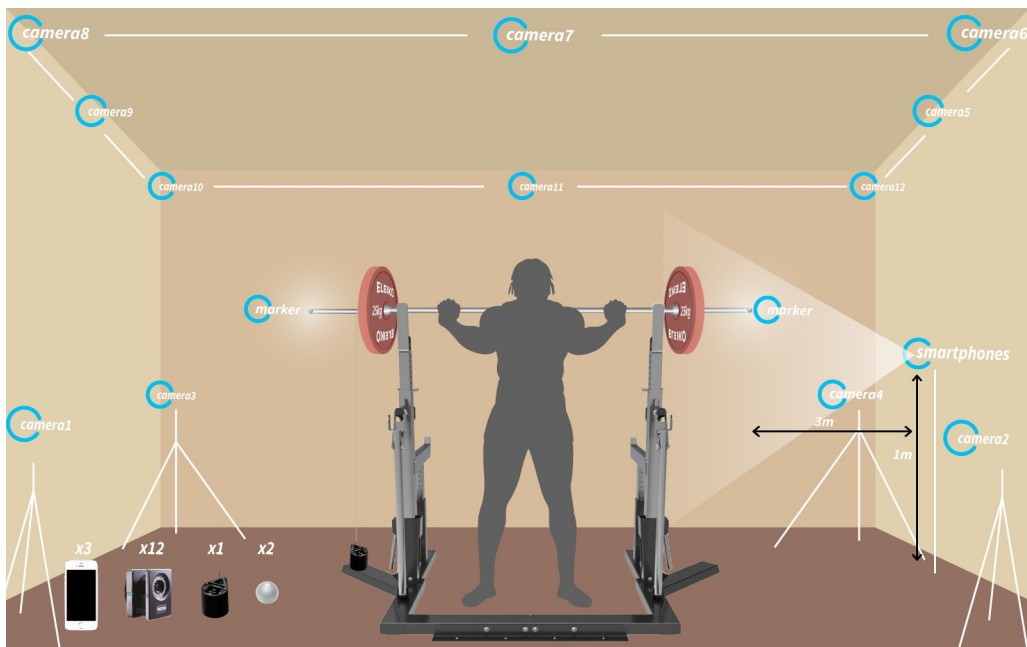

**Fig 1. Experimental setup.**

models with correlated random intercepts and slopes. For model 1, velocity output from the practical system was expressed as a function of velocity output from Vicon. For model 2, velocity output for each repetition was modeled as a function of source, where source was expressed as a binary dummy variable representing either Vicon or the practical system. Models were fitted according to the Bayesian framework, utilizing the rstan R package (version 2.26.23) and customized Stan scripts (Stan version 2.26.3). Weakly informative standardized priors were defined for all parameters. Based on model 1, estimated intercepts and slopes were interpreted concurrently as indicators of proportional bias, $R^2$ was calculated as a standardized indicator of random error, and the root mean squared error (RMSE) was calculated from posterior predictive distributions to represent absolute precision. Moreover, the standardized mean bias (SMB) was estimated from model 2. Posterior distributions were summarized using their mean and 95% Highest Density Interval (HDI).

The practical relevance of estimated error in smartphone-based velocity monitoring was evaluated using the models comparing Vicon to RO as a practical reference. In particular, the more extreme 95% HDI limit of each calculated statistic was used to construct a region of practical equivalence (ROPE) around the values indicating a perfect match, which are defined as follows: intercept = 0, slope = 1, $R^2$ = 1, RMSE = 0, SMB = 0. If the effect direction of the calculated statistic was supported by at least 95% of posterior probability, it was deemed statistically clear and labeled as "likely".

R and Stan scripts used for the analysis were uploaded to a public repository, to provide further details on the statistical modeling and applied priors (https://doi.org/10.5281/zenodo.10864547).

## Results

Out of the 600 planned repetitions of the 20 study participants, one squat repetition was not completed due to the weight being deemed too heavy. Furthermore, 8 deadlift repetitions were

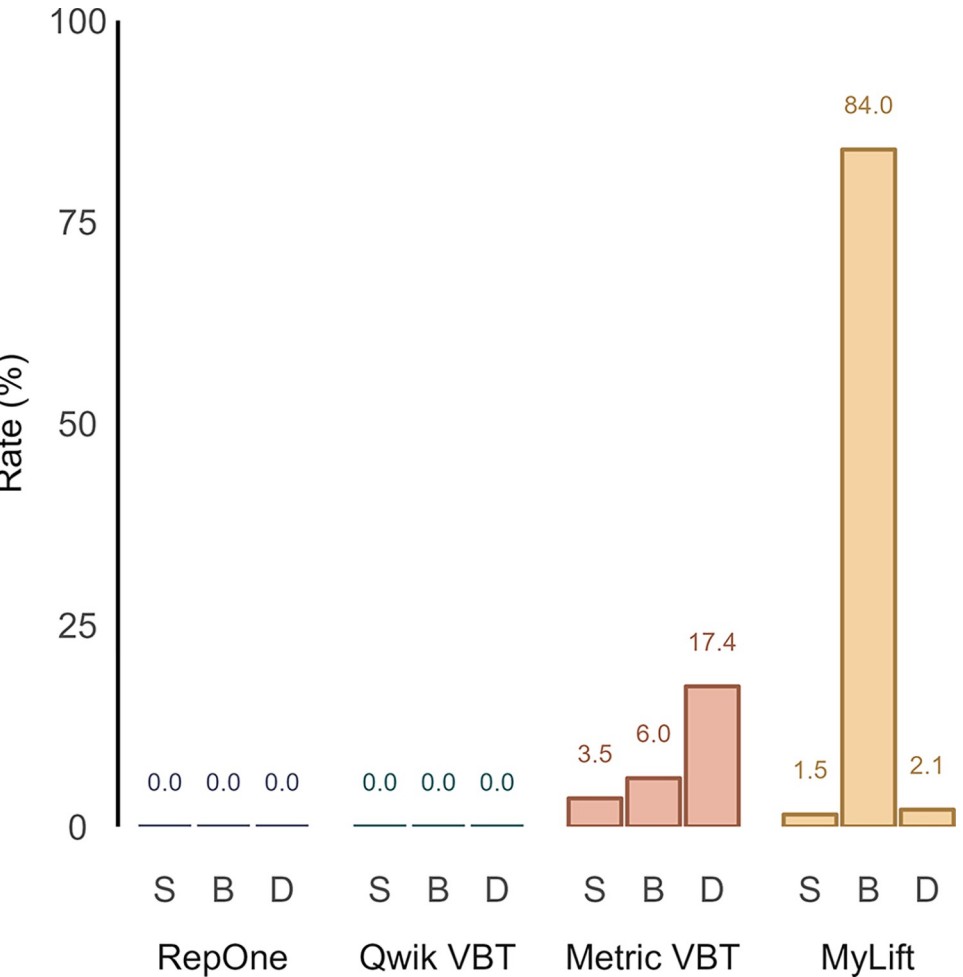

**Fig 2. Missed repetition rate across systems.** Values indicate the percentage of repetitions that were not identified by the system in the given exercise. B, bench press; D, deadlift; S, squat.

not performed for the same reason. One deadlift repetition was attempted but not completed due to the lifter losing grip of the barbell during the repetition and for one deadlift repetition the trajectories of the reflective pearl markers were not analyzable due to reflections of the barbell interfering with the markers. Fig 2 presents the missed repetition rates for the different systems and exercises. RO and QW showed perfect performance with no missed or ghost repetitions. ML, on the other hand, exhibited no ghost repetitions but had three and four missed repetitions in the squat and deadlift, respectively. Additionally, ML failed to record 168 reps in the bench press. Due to the low number of recorded repetitions, caution is advised when interpreting statistical analyses for ML in the bench press. MT was the only system to register ghost repetitions, totaling 16 (2 in the squat and 14 in the bench press). MT also displayed 7, 12, and 33 missed repetitions in the squat, bench press, and deadlift, respectively.

Figs 3–5 display the SMB, RMSE and $R^2$ calculated for each system-exercise combination.

For all investigated exercises, intercepts and slopes estimated from model 1 were likely equivalent between RO and QW, except for the deadlift (slope: p($\Theta \in$ ROPE) = 94.5%). Compared to RO, MT resulted in likely decreased slopes for all exercises (p($\Theta \in$ ROPE) < 2.5%), and likely increased intercepts for the bench press and deadlift (p($\Theta \in$ ROPE) < 0.1%). For ML, results were unclear with respect to ROPE in all cases. However, the intercept was likely

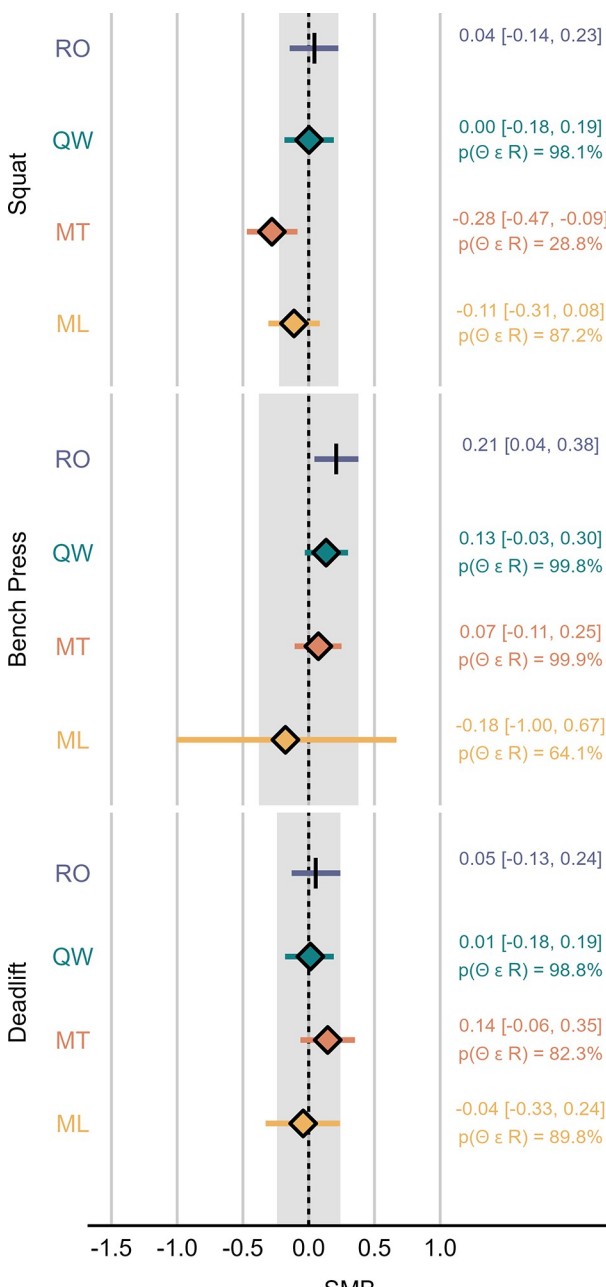

**Fig 3. Standardized mean bias (SMB) estimated from model 2.** The mean and 95% HDI of the posterior distribution for each device are summarized numerically to the right-hand side of the plot. Shaded areas mark the region of practical equivalence defined by the 95% Highest Density Interval (HDI) of RepOne. p(Θ ∈ R), probability of the parameter falling into the region of practical equivalence; ML, MyLift; MT, Metric VBT; QW, Qwik VBT; RO, RepOne.

below the line of identity in the squat (p(Θ < 0) = 99.7%) and likely above the line of identity in the deadlift (p(Θ > 0) = 98%). Moreover, the slope was likely lower compared to the line of identity in the deadlift (p(Θ < 1) = 95.6%). Fig 6 displays the group-level fits for model 1 with their respective intercept and slope.

The complete dataset which was used for all calculations was uploaded to a public repository (https://doi.org/10.5281/zenodo.10864547).

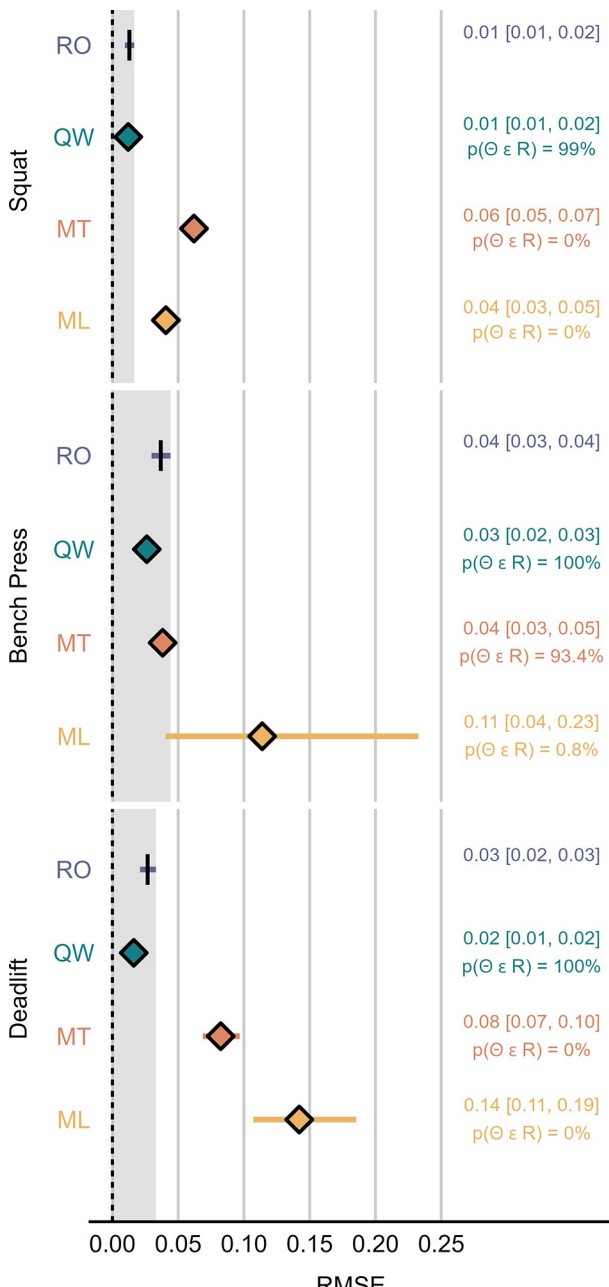

**Fig 4. Root mean square error (RMSE) of posterior predictive distributions calculated from model 1.** The mean and 95% HDI of the posterior distribution for each device are summarized numerically to the right-hand side of the plot. Shaded areas mark the region of practical equivalence defined by the 95% Highest Density Interval (HDI) of RepOne. p(Θ ϵ R), probability of the parameter falling into the region of practical equivalence; ML, MyLift; MT, Metric VBT; QW, Qwik VBT; RO, RepOne.

## Discussion

The goal of the present investigation was to determine the validity of 3 SAs and one LT determining mean barbell movement velocity across a range of loads in the powerlifting exercises. To the best of the authors' knowledge, this study is the first one to investigate the validity of multiple SAs and the RO.

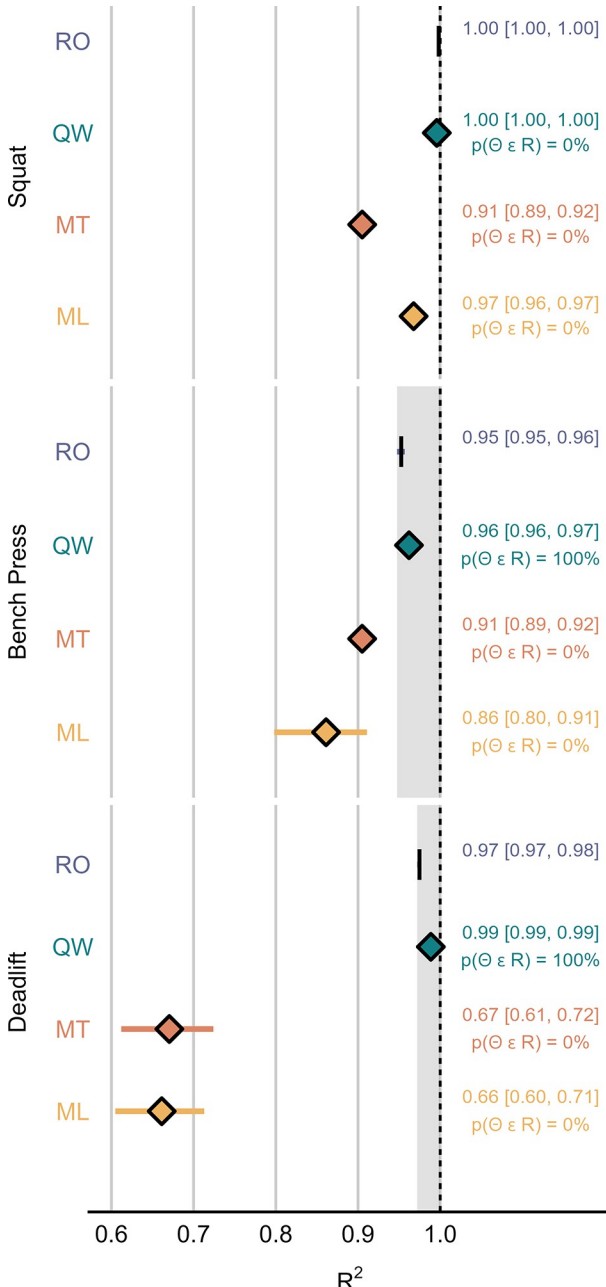

**Fig 5. Coefficient of determination ($R^2$) of model 1.** The mean and 95% HDI of the posterior distribution for each device are summarized numerically to the right-hand side of the plot. Shaded areas mark the region of practical equivalence defined by the 95% Highest Density Interval (HDI) of RepOne. p($\Theta \epsilon$ R), probability of the parameter falling into the region of practical equivalence; ML, MyLift; MT, Metric VBT; QW, Qwik VBT; RO, RepOne.

RO served as a practical criterion in the present study, since recent research promoted LTs as favorable field-based systems for velocity monitoring [4]. Our results indicate that RO is only subject to neglectable bias in the squat and deadlift, as portrayed by the SMB, with no indications of substantial over- or underestimation. These results conform with what has previously been reported for other LT systems, such as the GymAware Power Tool (GA). For example, Thompson et al. declared GA a valid and reliable measurement tool for mean velocity

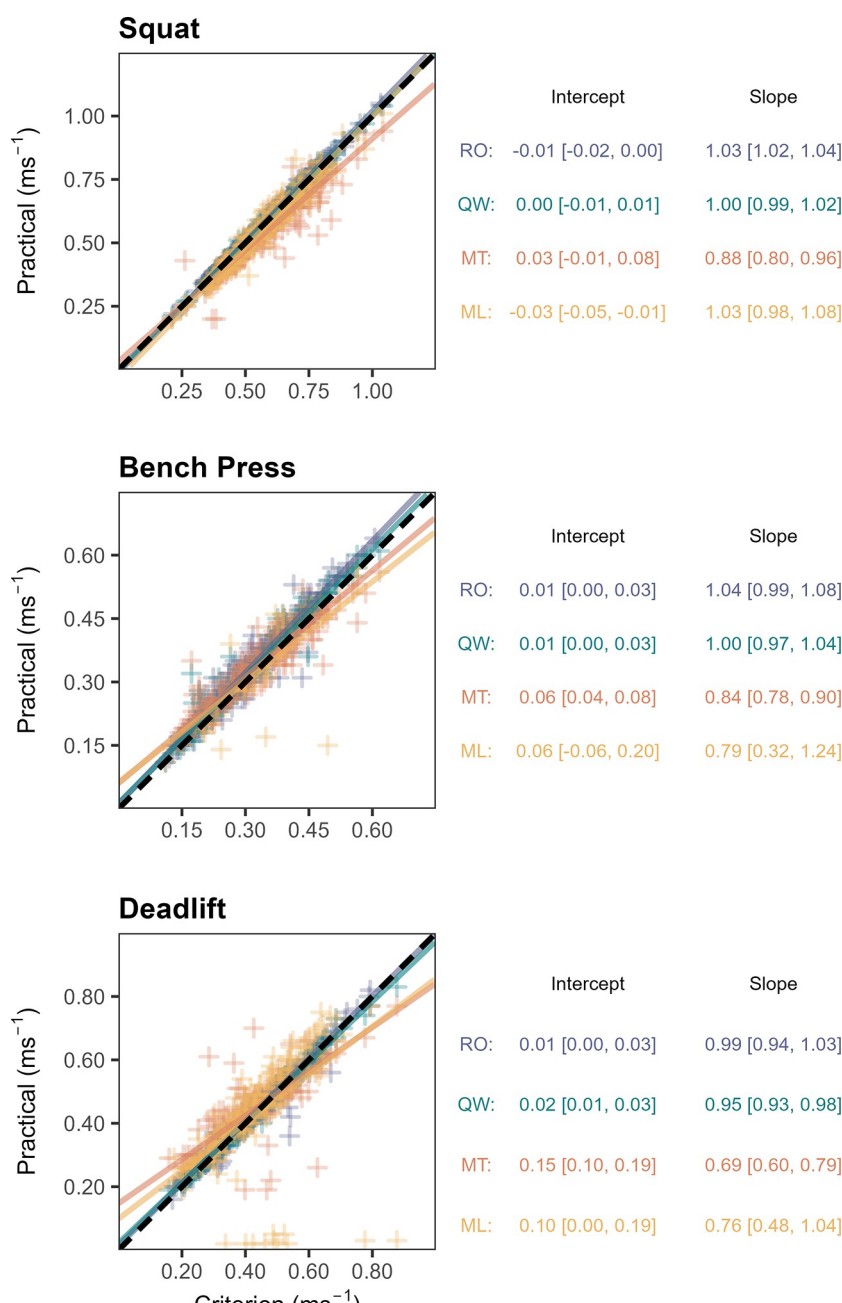

**Fig 6. Scatter plots with group-level functions of model 1.** For each practical system and investigated exercise, the mean and 95% HDI of the group-level intercept and slope are summarized numerically to the right-hand side of the plot. Dashed lines represent the line of identity (intercept = 0, slope = 1); ML, MyLift; MT, Metric VBT; QW, Qwik VBT; RO, RepOne.

in the back squat, where it performed exceptionally well in regression analyses (point estimate [95% confidence interval], intercept = -0.005 [-0.014, 0.003], slope = 0.991 [0.979, 1.004], $R^2$ = 0.99) [30]. Contrary to the squat and deadlift, we found that RO likely overestimates mean velocity in the bench press at trivial to small magnitude (mean [95% HDI], SMB = 0.21 [0.04, 0.38], mean difference = 0.02 [0, 0.04] m·s$^{-1}$). Model functions further illustrate a slight tendency for proportional bias in the bench press, where error increases with movement velocity.

While an exploratory analysis of factors causing the exercise-specific bias in RO was beyond the scope of the present study, it may be assumed that the effect could be partially explained by kinematic characteristics of the investigated exercises, such as horizontal barbell displacement during the concentric movement phase. Importantly, a study conducted by Mitter et al. under similar experimental conditions showed that GA may perform at slightly higher standardized accuracy in the bench press [19]. In the present study, random error of RO, as evaluated by $R^2$ and RMSE, was also more pronounced in the bench press compared to the other two exercises. Our findings on RMSE in RO conform to what previously mentioned research reported on GA [19], suggesting that RO might perform at comparable precision. Overall, readers should be aware that for the bench press, RO may not be universally considered an ideal practical criterion to evaluate bias in SAs. Thus, findings from respective ROPE-analyses should be interpreted with caution.

QW yielded a level of validity that was equivalent to RO for all investigated statistics and exercises, except for $R^2$ in the squat, where more than 99.9% of posterior probability fell below the ROPE threshold defined by RO. Importantly, QW still showed excellent validity on its own and the lack of overlapping posterior distributions may be explained by the HDIs being exceptionally narrow in both cases (mean [95% HDI], RO: $R^2$ = 0.9983 [0.9981, 0.9984], QW: $R^2$ = 0.9959 [0.9955, 0.9962]). Based on the results of model 1, there was a 94.5% probability of the slope parameter falling into the ROPE, thus missing the predefined threshold for clear effects. Interestingly, in the bench press and deadlift, there was a tendency for QW to provide higher precision compared to RO, as evidenced by a slightly lower RMSE. Among the three investigated SAs, QW was the only one to record all completed repetitions. No ghost repetitions were recorded by QW during data acquisition.

SMB estimates for MT indicate that, on average, the application provides equivalent accuracy compared to RO in the bench press. Results were deemed unclear in the other two exercises. However, the presence of proportional bias was identified based on likely substantial deviations of the intercept and slope parameters in the bench press and deadlift, as well as for the slope parameter in the squat. In all cases, statistics reveal a tendency of MT underestimating velocity at higher barbell speed. These trends were paired with a tendency of MT overestimating velocity at lower barbell speed in the bench press and deadlift. In terms of random error, $R^2$ statistics highlight worse performance compared to RO that was likely substantial in all investigated exercises. Similarly, precision was substantially worse compared to RO in the squat and deadlift, as illustrated by the RMSE. In case of the bench press, data hint towards equivalent precision when comparing MT to RO, with a 93.4% probability of the RMSE falling into the ROPE. Regarding its ability to correctly detect repetitions across powerlifting exercises, MT was the only investigated smartphone application that recorded ghost repetitions during data acquisition. Furthermore, it failed to record single repetitions in all three exercises, with its missed repetition rate being highest in the deadlift. To our knowledge, MT has thus far only been validated in one more peer-reviewed study, which compared it to a gold standard 3D Motion Capture System and reported Pearson's correlation coefficients ranging from 0.67 to 0.95 across different exercise categories [31]. While the authors declared MT as a valid and reliable measurement method for the assessment of mean velocity, it was not directly compared to a LT. However, the accuracy levels reported for MT were still well below the accuracy levels typically seen in LTs [10, 19]. This is further corroborated by the findings of the present investigation. Moreover, the creators of MT published an internal validation study, where they compared MT with a gold standard 3D high-speed motion capture system [32]. Similar to the present study, a slight tendency for MT to underestimate mean velocity in the front squat and at high movement speed in the deadlift, but only neglectable mean bias in the bench press was found.

Data collected from ML did not reveal clear estimates for SMB, but a tendency towards trivial bias in the squat and deadlift. No clear signs of proportional bias were identified for ML in the squat and bench press. However, analysis indicated the presence of proportional bias in the deadlift, resulting in a tendency of ML underestimating velocity at higher barbell speed. This effect may to some extent be caused by multiple outliers produced by four participants at loads ranging from 45% to 75% 1RM, for which ML recorded exceptionally low movement velocity of less than $0.06 \text{m} \cdot \text{s}^{-1}$. Thus, the identified proportional bias may not validly apply to most observations, and extreme outliers may be explained by confounders other than movement velocity. In terms of random error, as portrayed by $R^2$, and precision, as portrayed by RMSE, ML performed substantially worse compared to RO in all three exercises. While ML missed single repetitions during data acquisition in all three exercises, this was only observed at a low rate in the squat and deadlift. In the bench press, however, ML registered only 16% of completed repetitions. It may be assumed that ML's ability to successfully detect repetitions is inversely associated with characteristic features of the bench press, such as the distance traveled by the barbell in horizontal and vertical direction. Readers should be aware that the comparably low sample size drawn from ML in the bench press may compromise the robustness of derived statics. These results are in accordance with the findings of Martínez-Cava et al. who, although only examining PV, also declared ML less accurate than a LT [17].

Overall, the results of the present study indicate that among the three investigated SAs, QW is the only one recording mean velocity at an accuracy and precision comparable to RO in all three powerlifting exercises.

In general, it seems like the kinematic thresholds used by the different developers play a major role in the validity of the different SAs. If the threshold for range of motion is set too low, more ghost repetitions will be detected. If it is set too high, more missed repetitions will be the consequence. If the threshold for instantaneous velocity to identify the start and end of the concentric movement phase is too low, different movement and/or movement phases beyond the concentric phase of the actual repetitions might be included in a measurement. If it is set too high, parts of the repetition might be excluded as a consequence. It seems like QW, along with its features to tag the plates and the option to cut the native video to a length where only the repetition is included, selected the best kinematic thresholds for valid mean velocity measurements in powerlifting exercises.

Despite using an approved gold standard criterion for kinematic measurement, the results of this investigation should be treated with caution. This is primarily due to two factors. First, a validation study can only supply momentary insight into a dynamically changing system. Developers of SAs typically publish software updates on a regular basis. Therefore, the results of the present investigation may not speak for more recent software versions of each analyzed system. Second, the study design yields only limited external validity. All SAs were tested under favorable conditions. A standardized filming position with the smartphones placed in a stable stand was used. Furthermore, the area of the laboratory where data collection was completed was free of any objects which might disturb the measurement. Adequate lighting was also ensured. Therefore, it might be difficult to predict how the investigated systems will perform in a regular training environment. This issue seems to be more relevant for the SAs than the RO since the RO only involves placing the unit on a magnetic surface and attaching the wire of the unit with a strap onto the barbell, in a way to minimize horizontal displacement. This represents a closed system which is much less reliant on other external factors. The usage of SAs, on the other hand, may potentially be limited by a multitude of confounders introduced by a regular training environment. Such confounders may include the suboptimal placement and orientation of the smartphone's camera, or interferences caused by other objects and people, affecting recordings and lighting.

The findings of this study provide evidence that SAs can be a valid method of assessing mean concentric barbell velocity in powerlifting exercises. Using the same hardware, different SAs delivered vastly heterogeneous results. The authors recommend exercising caution when using non-validated SAs to measure mean concentric barbell velocity in powerlifting exercises, as the SAs may be prone to substantial measurement error and low repetition detection rates. Among the SAs tested in this study, the Qwik VBT system delivered the best results, showing high levels of accuracy and precision across exercises. Notably, it performed as well as, if not better than, the RepOne LT system and may, thus, contribute to overcoming the financial barrier to the widespread application of velocity-based training methodology. However, it should be noted that the study identified more limitations than advantages for SAs overall, and the Qwik VBT system stands out as an exception rather than the rule.

## Acknowledgments

The authors express their gratitude to all athletes who participated in this study and further to everyone who gave up their time to make this investigation possible. The authors had no professional relationship with any companies or manufacturers of products validated in this study. There are no conflicts of interest related to the content of this study.

## Author Contributions

**Conceptualization:** Alexander Renner, Arnold Baca.

**Data curation:** Alexander Renner.

**Formal analysis:** Alexander Renner, Benedikt Mitter, Arnold Baca.

**Investigation:** Alexander Renner.

**Methodology:** Alexander Renner.

**Project administration:** Arnold Baca.

**Software:** Alexander Renner, Benedikt Mitter.

**Supervision:** Benedikt Mitter, Arnold Baca.

**Validation:** Alexander Renner.

**Visualization:** Alexander Renner, Benedikt Mitter.

**Writing – original draft:** Alexander Renner, Benedikt Mitter.

**Writing – review & editing:** Alexander Renner, Benedikt Mitter, Arnold Baca.

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
