## [Decision Letter · Decision Letter 0]

28 Aug 2024

PONE-D-24-27051Concurrent validity of novel smartphone-based apps monitoring barbell velocity in powerlifting exercisesPLOS ONE

Dear Dr. Renner,

Thank you for submitting your manuscript to PLOS ONE. After careful consideration, we feel that it has merit but does not fully meet PLOS ONE’s publication criteria as it currently stands. Therefore, we invite you to submit a revised version of the manuscript that addresses the points raised during the review process.

We look forward to receiving your revised manuscript.

Kind regards,

Gianpiero Greco

Academic Editor

PLOS ONE

Journal requirements: 1. When submitting your revision, we need you to address these additional requirements. Please ensure that your manuscript meets PLOS ONE's style requirements, including those for file naming. The PLOS ONE style templates can be found at https://journals.plos.org/plosone/s/file?id=wjVg/PLOSOne_formatting_sample_main_body.pdf and https://journals.plos.org/plosone/s/file?id=ba62/PLOSOne_formatting_sample_title_authors_affiliations.pdf.

Reviewers' comments:

Reviewer's Responses to Questions

**Comments to the Author**

1. Is the manuscript technically sound, and do the data support the conclusions?

Reviewer #1: Yes

Reviewer #2: Yes

2. Has the statistical analysis been performed appropriately and rigorously? 

Reviewer #1: Yes

Reviewer #2: No

3. Have the authors made all data underlying the findings in their manuscript fully available?

Reviewer #1: Yes

Reviewer #2: Yes

4. Is the manuscript presented in an intelligible fashion and written in standard English?

Reviewer #1: Yes

Reviewer #2: Yes

5. Review Comments to the Author

Reviewer #1: The article was properly prepared in accordance with scientific research standards. The authors used appropriate research methodology. The reservations that may arise concern only a small number of sources used in the bibliography (it would be worth supplementing it in subsequent articles with publications by Prof. Michał Wilk or Prof. Michał Krzysztofik). The topic is extremely interesting due to the usefulness of the results achieved in the process of sports training or training of people practicing strength sports recreationally.

Reviewer #2: Review of “Concurrent validity of novel smartphone-based apps monitoring barbell velocity in powerlifting exercises”

This paper presents a valuable effort to examine the validity of three smartphone apps designed to quantify the velocity of a barbell during the three major powerlifting exercises. The study’s approach, which includes the use of a linear position transducer and a comparison with the Vicon system as the gold standard, is commendable. Additionally, it’s great to see that the study included both male and female powerlifters, providing a more comprehensive analysis.

Abstract:

To strengthen the abstract, it would be beneficial to include data on the typical mean velocity or the relative size of the RMSE compared to the mean velocity. This additional information would help readers quickly grasp the significance of the findings.

Introduction:

While the introduction provides a solid foundation, the use of abbreviations could be made clearer. For instance, “MV” stands for mean velocity, but “MT” refers to an app, which may be confusing for readers.

•Line 42: The discussion around the different velocity parameters (mean velocity, peak velocity, etc.) could be elaborated to better reflect the ongoing debate in the field.

•Line 54: It might be worth mentioning that there are also smartwatch applications available that serve similar functions.

Table 1:

The values in Table 1 would benefit from rounding to more meaningful figures. For example, 541.88±155.91 could be rounded to 541±156 to enhance readability and comprehension.

Methodology:

•Line 169: A more detailed description of the barbell fixation would be helpful. Including a picture of the setup would significantly improve clarity.

•Additionally, please provide information on the frequency of the sensors used in the iPhone.

•Line 207: Clarifying the thresholds and any other steps involved in data processing would enhance the transparency of the methodology.

Figures:

The resolution of the figures, particularly Figure 6, needs improvement as it is currently difficult to read. High-quality visuals are crucial for effective communication of the data.

Results Section:

The beginning of the results section currently reads more like a continuation of the methods. Revising this to make it more reader-friendly would improve the flow of the paper.

•Line 248: Further clarification is needed here. Could you explain the reasoning behind this choice?

•Text and Figure 2: It would be helpful to present the data only once to avoid redundancy. Including ghost repetitions in Figure 2 would also provide a clearer representation of the findings.

Currently, the results section blends text and figure legends. If the legends are removed, there is little information left in the text. Including more detailed information directly in the results section would enhance its substance.

Discussion:

Lastly, the addition of two models to interpret the data seems to introduce some confusion. It would be beneficial to ensure that these models align closely with the study’s objectives, or to reconsider their inclusion if they do not add substantial value.

I hope these suggestions help in refining your manuscript. With these revisions, the paper has the potential to make a significant contribution to the field.

6. PLOS authors have the option to publish the peer review history of their article (what does this mean?). If published, this will include your full peer review and any attached files.

Reviewer #1: No

Reviewer #2: **Yes: **Silvio Lorenzetti

---

## [Author Response · Author response to Decision Letter 0]

9 Oct 2024

Dear reviewers, 

we would like to thank you for the constructive feedback. You can find our detailed response in the document "Response to Reviewers".

---

## [Decision Letter · Decision Letter 1]

4 Nov 2024

Concurrent validity of novel smartphone-based apps monitoring barbell velocity in powerlifting exercises

PONE-D-24-27051R1

Dear Dr. Renner,

We’re pleased to inform you that your manuscript has been judged scientifically suitable for publication and will be formally accepted for publication once it meets all outstanding technical requirements.

Kind regards,

Gianpiero Greco

Academic Editor

PLOS ONE

Additional Editor Comments (optional):

Reviewers' comments:

Reviewer's Responses to Questions

**Comments to the Author**

1. If the authors have adequately addressed your comments raised in a previous round of review and you feel that this manuscript is now acceptable for publication, you may indicate that here to bypass the “Comments to the Author” section, enter your conflict of interest statement in the “Confidential to Editor” section, and submit your "Accept" recommendation.

Reviewer #2: All comments have been addressed

2. Is the manuscript technically sound, and do the data support the conclusions?

Reviewer #2: Yes

3. Has the statistical analysis been performed appropriately and rigorously? 

Reviewer #2: Yes

4. Have the authors made all data underlying the findings in their manuscript fully available?

Reviewer #2: Yes

5. Is the manuscript presented in an intelligible fashion and written in standard English?

Reviewer #2: Yes

6. Review Comments to the Author

Reviewer #2: Thank you for the revision. All the raised points have been addressed properly. It is fine for me and the letter was helpful.

7. PLOS authors have the option to publish the peer review history of their article (what does this mean?). If published, this will include your full peer review and any attached files.

Reviewer #2: **Yes: **Silvio R. Lorenzetti

---

## [Editor Report · Acceptance letter]

8 Nov 2024

PONE-D-24-27051R1 

PLOS ONE

Dear Dr. Renner, 

I'm pleased to inform you that your manuscript has been deemed suitable for publication in PLOS ONE. Congratulations! Your manuscript is now being handed over to our production team.

Kind regards, 

on behalf of

Dr. Gianpiero Greco 

Academic Editor

PLOS ONE